# SARS-CoV-2 Reinfection and Severity of the Disease: A Systematic Review and Meta-Analysis

**DOI:** 10.3390/v15040967

**Published:** 2023-04-14

**Authors:** Nhu Ngoc Nguyen, Y Ngoc Nguyen, Van Thuan Hoang, Matthieu Million, Philippe Gautret

**Affiliations:** 1Aix Marseille Univ, IRD, AP-HM, SSA, VITROME, Marseille, France; 2IHU-Méditerranée Infection, Marseille, France; 3Hoan My Sai Gon Hospital, Ho Chi Minh City, Vietnam; 4Thai Binh University of Medicine and Pharmacy, Thai Binh, Vietnam; 5Aix Marseille Univ, IRD, AP-HM, MEPHI, Marseille, France

**Keywords:** COVID-19, SARS-CoV-2, coronavirus, reinfection, second infection, variant

## Abstract

Since the discovery of SARS-CoV-2, changes in genotype and reinfection with different variants have been observed in COVID-19-recovered patients, raising questions around the clinical pattern and severity of primary infection and reinfection. In this systematic review, we summarize the results of 23 studies addressing SARS-CoV-2 reinfections. A total of 23,231 reinfected patients were included, with pooled estimated reinfection rates ranging from 0.1 to 6.8%. Reinfections were more prevalent during the Omicron variant period. The mean age of reinfected patients was 38.0 ± 6. years and females were predominant among reinfected patients (M/F = 0.8). The most common symptoms during the first and second infection were fever (41.1%), cough (35.7% and 44.6%), myalgia (34.5% and 33.3%), fatigue (23.8% and 25.6%), and headaches (24.4% and 21.4%). No significant differences of clinical pattern were observed between primary infection and reinfection. No significant differences in the severity of infection were observed between primary infection and reinfection. Being female, being a patient with comorbidities, lacking anti-nucleocapsid IgG after the first infection, being infected during the Delta and Omicron wave, and being unvaccinated were associated with a higher risk of reinfection. Conflicting age-related findings were found in two studies. Reinfection with SARS-CoV-2 suggests that natural immunity is not long-lasting in COVID-19 patients.

## 1. Introduction

Severe acute respiratory syndrome coronavirus-2 (SARS-CoV-2) is the virus responsible for the infection known as coronavirus disease 2019 (COVID-19) [1]. Reinfection may occur after recovery from the disease. The first published case of SARS-CoV-2 reinfection was documented in a 33-year-old Hong Kong man in August 2020, with two distinct virus strains corresponding to two episodes of infection [2]. In a meta-analysis addressing the risk of SARS-CoV-2 reinfection in discharged patients, the pooled estimated reinfection rate was 0.3% [3]. However, this study was conducted before the Omicron variant wave and did not compare clinical symptoms during the first and second infections. We recently observed that the Omicron variant was associated with an increased risk of reinfection in recovered patients [4]. The length of time between two episodes used to define a SARS-CoV-2 reinfection varies. The US Centers for Disease Control and Prevention uses the time frame of 90 days [5], while the European Centre for Disease Prevention and Control uses 60 days or more [6]. However, a shorter duration has been reported in unvaccinated patients, with a mean duration of 47 days (range 17–65 days) between two episodes of infection [7]. The risk factors for SARS-CoV-2 reinfection with recent variants of concern following a previous infection remain unclear. Therefore, we conducted an updated systematic review and meta-analysis aiming at assessing the proportion of reinfections with SARS-CoV-2 over time. We also compared the clinical patterns and severity of infection in reinfected patients to their status during primary infection. Finally, we aimed to identify risk factors for reinfection.

## 2. Materials and Methods

### 2.1. Protocol and Search Strategy

The Preferred Reporting Items for Systematic Review and Meta-Analysis (PRISMA) guidelines were applied to perform this systematic review (http://www.prisma-sattement.org, accessed on 31 March 2023). The PubMed (http://www.ncbi.mlm.nih.gov/pubmed, accessed on 31 March 2023) and Google Scholar (http://scholar.google.fr, accessed on 31 March 2023) databases were investigated to identify all relevant studies. We used an expanded MeSH to identify a maximum number of keyword synonyms. The current search was performed on 31 March 2023 by combining the following keywords:

#1: “COVID-19” OR “SARS-CoV-2”

#2: “Re-infection?” OR “Reinfection?” OR “Recurrent Infection” OR “Re-positive”

#3: #1 AND #2

### 2.2. Eligibility Criteria

All published studies and pre-prints (including case series and prospective cohort studies) were selected according to the following inclusion criteria:

(1) Studies reporting on adult COVID-19 patients (≥18 years old) with infection confirmed by a positive RT-PCR (Reserve Transcription Polymerase Chain Reaction);

(2) Due to the heterogeneity of duration between two episodes of infection, we included studies reporting on patients that met the following criteria, based on the US CDC Common Investigation Protocol for Investigating Suspected SARS-CoV-2 Reinfection [8]:

+ Two episodes of infection at ≥90 days interval in patients with negative RT-PCR results between two episodes of infection, including asymptomatic patients;

+ Two episodes of infection at 60 to 89 days of interval: symptomatic patients confirmed by RT-PCR test with ≥ one intermediate negative RT-PCR test with no other reason or symptomatic patients tested with different genotypes between two episodes of infection;

+ Two episodes of infection at an interval of <60 days: symptomatic patients confirmed by differences between two episodes of infection in whole genome sequencing in patients infected with different virus genotypes;

Only papers in English were included in this work.

### 2.3. Exclusion Criteria

We excluded single-case reports and all studies based on animal experiments. Review papers were excluded, though the reference lists were assessed to identify potentially relevant studies. We also excluded studies conducted only on immunocompromised patients with cancer and acquired immunodeficiency syndrome.

### 2.4. Data Collection Process

Four researchers independently checked the relevant articles and data extraction. All discordant results were discussed to reach a consensus between researchers.

The type of study, country, and setting where studies were performed, the number and proportion of reinfection cases, demographic information (age and sex), comorbidities and vaccination status, the time between two episodes of infection, symptoms, imaging findings, and severity outcomes (hospitalization and transfer to intensive care unit and death) were extracted from the included articles. When available, risk factors for reinfection were reported or calculated.

### 2.5. Assessment for Quality of Studies

The quality of cohort studies was accessed using the Newcastle–Ottawa Scale (NOS) [9], consisting of eight items with a maximum score of nine. The score of the study classifies the quality as good (7–9), fair (4–6), and poor (0–3). The quality of the cases series was evaluated using the National Institute for Health and Clinical Excellence (NICE) checklist, allowing the following stratification of series: good quality (score ≥ 7), fair quality (score 4–6), and poor quality (score 0–3) [10].

### 2.6. Statistics

All statistical analyses were performed using the open-source software R [R Core Team] R: A language and environment for statistical computing from the R foundation for statistical computing, based in Vienna, Austria, in 2020. URL: [http://www.r-project.org, accessed on 31 March 2023]. In the present study, the heterogeneity I2 > 50% was identified as representing substantial heterogeneity [11]. The pooled effect size and 95% confidence interval (95% CI) were given using the random effect model. A funnel plot was applied to estimate publication bias. We excluded all studies from the analysis where there was no event in either arm, according to the Cochran guide [12].

The most frequent variant circulating at the time of reinfection in each country was assessed using the CoVariants website (https://covariants.org/, accessed on 31 March 2023) and, when available, the SARS-CoV-2 sequencing data described in each study.

## 3. Results

### 3.1. Study Selection and Quality Assessment

Figure 1 shows the search algorithm. A total of 3051 articles were identified from PubMed and Google Scholar. After removing duplicates, 2903 articles were scanned by reading the title and abstract. A total of 237 articles were accessed for the full-text screening. Ultimately, 26 articles met the inclusion criteria for the meta-analysis (Appendix A).

Studies assessed using the NICE criteria were all scored as good quality (eight studies) and those assessed with NOS were scored as either good quality (seven studies) or fair quality (eleven studies) (Appendix A).

### 3.2. Characteristics of Selected Studies

Of the 26 included studies, eight were case series [13,14,15,16,17,18,19,20], 13 were prospective cohort studies [21,22,23,24,25,26,27,28,29,30,31,32,33], and five were retrospective cohort studies [34,35,36,37,38]. Six studies were conducted on healthcare workers (HCWs) [13,16,17,25,33,37], one was conducted on a majority of HCWs (281/284 patients) [32], and one was conducted on young student athletes [30]. Most studies were conducted in Europe (Appendix A), including two in France [26,28], two in the United Kingdom (UK) [23,24], one in Italy [34], one in Denmark [29], one in Portugal [20], and one in Austria [36], followed by the Middle-East (six studies), with two in Qatar [21,27], one in Iran [15], one in Iraq [22], one in Israel [35], and one in Saudi Arabia [38]. Four studies were conducted in Brazil [16,17,18,32] and four in the United States [25,30,31,33]. In addition, three studies were conducted in Asia, including two in India [13,37] and one in South Korea [14]. Finally, one study was conducted in Africa; this study took place in Gambia [19]. Most studies were monocentric (n = 13) [13,15,16,17,18,19,20,22,26,28,31,37,38], while some were multicentric (n = 8) [14,23,25,29,30,33,34,35] or national (n = 5) [21,24,27,32,36].

### 3.3. SARS-CoV-2 Variants

According to the CoVariants website, the most frequent variant SARS-CoV-2 circulating at the time of reinfection in countries where studies were conducted were Alpha variant in four studies [23,24,31,35], Delta variant in four studies [26,30,37,38], Omicron variant in two studies [27,33], 20A.EU2 variant in two studies [28,36], 20E.EU1 variant in one study [22], 20 H (Beta) variant in one study [21], and 21C (Epsilon) variant in one study [25] (Appendix A).

In ten studies, SARS-CoV-2 variants were identified by genetic methods for 2354 reinfected patients [17,18,19,21,23,26,27,31,33,38]. The most frequent variants were Delta (40.1%), followed by Omicron (21.9%) and B.1.351 (Beta) variant (19.6%) (Appendix A).

### 3.4. Prevalence of Reinfection and Characteristics of SARS-CoV-2 Reinfected Patients

#### 3.4.1. Prevalence of Reinfection and Effect of SARS-CoV-2 Variants and Region

A total of 23,231 reinfected patients were included, including 46 patients from the case series [13,14,15,16,17,18,19,20] and 23,185 patients identified from the cohort studies [21,22,23,24,25,26,27,28,29,30,31,32,33,34,35,36,37,38] (Appendix A). The numbers of patients in the case series ranged from two to 25 and the numbers of patients in the cohort studies from seven to 13,960 patients. A denominator (discharged patients) was available in 18 studies (n = 5,080,830) [21,22,23,24,25,26,27,28,29,30,31,32,33,34,35,36,37,38], with reinfection rates ranging from 0.1% to 6.8%. The funnel plot shows marked asymmetry, with most studies on the right side of the effect being estimates from individual studies (Appendix A). To assess the potential reasons for this heterogeneity, we assessed the proportion of reinfection per region in 18 cohort studies [21,22,23,24,25,26,27,28,29,30,31,32,33,34,35,36,37,38]. The pooled proportion of reinfection by region was relatively similar with 2.1%, 0.8%, and 0.7% for studies conducted, respectively, in the Americas (US and Brazil) [25,30,31,32,33], the Middle-East [21,22,27,35,38] and Europe [23,24,26,28,29,34,36] (Appendix A). By contrast, in one study conducted in India among health care workers, the prevalence of reinfection was 6.1% [37]. In each region, a high level of heterogeneity was observed between studies. We also separated studies according to the most frequent SARS-CoV-2 variant circulating at the time of reinfection (Appendix A). Three studies conducted over long periods of time, for which the identification of a dominant SARS-CoV-2 variant was not feasible, were excluded from this analysis [29,32,34]. The highest pooled proportion of reinfection was observed in patients reinfected with Omicron variant (4.4%) [27,33] in two studies, followed by Delta variant (1.1%) [26,30,37,38] in four studies, and Alpha variant (0.2%) in four studies [23,24,31,35]. In addition, a 1.1% pooled prevalence of reinfection was observed in patients infected by other SARS-CoV-2 variants (including 0.3–0.7% with 20A.EU2 in two studies [28,36], 3.1% with 20E.EU1 [22], 0.5% with 21H (Beta) [21], and 5.9% with 21C (Epsilon)) [25]. Once again, high heterogeneity was observed between studies conducted in patients reinfected with each variant. To further assess this heterogeneity, studies were divided into studies conducted during the pre-Omicron and Omicron periods. In addition, in the pre-Omicron studies, we separately analyzed the studies conducted on healthcare workers [25,37] and the general population [21,22,23,24,26,28,30,31,35,36,38]. Regarding studies conducted during the pre-Omicron period, the pooled proportion of reinfection was 0.5%, 95% CI = [0.3–0.6] in the general population and 6.0%, 95% CI [5.3–6.7] in healthcare workers (Appendix A). High heterogeneity was still observed in pre-Omicron studies, with a 3.1% reinfection rate in one study [22]; however, it ranged from 0.1% to 0.8% in other studies. We found no explanation for this heterogeneity. In contrast, the two studies conducted on healthcare workers showed very close reinfection rates of 5.9% and 6.1%. A pooled proportion of 4.4%, 95% CI [2.0–8.5%] reinfection rate was observed in the two studies conducted during the Omicron period, with a prevalence of 2.9% in the general population [27] and 6.8% in health care workers [33].

#### 3.4.2. Demographics

##### Age

The overall age of discharged patients from four cohort studies [21,24,25,34] with 3,988,233 discharged patients ranged from 20 to 55 years (Appendix A). The overall age from twenty studies with 21,588 reinfected patients ranged from 32 to 48 years [13,14,15,16,17,19,20,21,22,24,25,26,27,28,31,32,34,35,36,38]. In studies where comparison was possible between discharged and reinfected patients, no marked differences in age were observed in three studies [21,25,34], while the age of reinfected patients was significantly higher in another study [22].

##### Gender

The male/female ratio from six cohort studies with 3,991,528 discharged patients [21,24,25,30,34,37] was 1,880,775/2,110,752 (0.89 with 52.9% female patients) and gender was “other” in one patient (Appendix A). The male/female ratio was 9558/12,499 (0.8 with 60.0% female patients) from 24 studies with 22,057 reinfected patients [13,14,15,16,17,18,19,20,21,22,23,24,25,26,27,28,31,32,33,34,35,36,37,38]. In studies where a direct comparison was possible between discharged patients and reinfected patients, the proportion of females among the reinfected patients was slightly higher [21,25,34].

#### 3.4.3. Comorbidities

Only one study reported on detailed comorbidities for 119,226 discharged patients with hypertension (9.3%), cardiovascular disease (8.3%) and diabetes (4.2%) being the most frequent [34]. Comorbidities were detailed in 5873 reinfected patients in 14 studies [13,14,16,17,19,20,26,28,30,31,32,34,35,38] (Appendix A). The most common comorbidities were hypertension (4.4%), diabetes (4.3%), chronic respiratory diseases (3.3%), obesity (3.3%), and cardiovascular diseases (3.1%). In the study where a comparison was possible between discharged and reinfected patients, comorbidities were less prevalent in reinfected patients [34].

#### 3.4.4. Vaccination Status

One study reported information on COVID-19 vaccination before the first infection with a prevalence of 0/207 patients [26] (Appendix A). Vaccination status was reported in nine studies [17,18,24,26,30,32,33,34,38]. Among 19,982 patients, 7742 (40.0%) had received at least one dose of vaccine. The IgG positivity rate after the first infection was documented in eight studies [14,15,16,20,22,23,26,31]; 57.3% (55/96) of patients tested positive for IgG following the first infection, including anti-nucleocapsid IgG in one study [22] and anti-spike IgG in seven studies [14,15,16,20,23,26,31].

### 3.5. Length of Time between Two Episodes

The length of time between two infections was described in 19 studies, including 17 studies with a calculated mean ± SD of 178.9 ± 87.1 days, ranging from 31 to 647 days [13,14,15,16,17,18,19,20,25,26,28,30,31,32,34,35,36] (Appendix A). Two other studies reported a gap between two infections with a median time of 183 days [37] and 277 days [21], ranging from 131 to 351 days. Only one study [14] reported a duration between two episodes of infection of less than 90 days from confirmation of the genome sequencing result.

### 3.6. Clinical Symptoms

Seven studies reported detailed symptoms for both episodes of infection in 168 reinfected patients [15,16,17,20,22,23,26]. In one of these studies, we included 121/209 patients for whom symptom information was available during both episodes [26]. The most common symptoms during the first and second infection were fever (41.1%), cough (35.7% and 44.6%), myalgia (34.5% and 33.3%), fatigue (23.8% and 25.6%), and headaches (24.4% and 21.4%). The prevalence of other symptoms is presented in Figure 2. Although sore throats tended to be less frequent during the first infection as compared to the second infection, no significant differences in the prevalence of symptoms were observed.

Only one study [14] reported the results of computed tomography chest at first infection with abnormal (ground-glass opacity) in 2/4 patients. Two studies reported abnormal chest CT results [14,15] in 3/7 patients with a second infection.

### 3.7. Severity of Infection

A total of 12 studies reported information on the hospitalization rate for both episodes of infection in 14,136/14,460 patients [14,15,16,17,18,20,23,24,26,28,35,36]. We excluded all studies from the analysis where there was no event of hospitalization in either arm. As a result, 14,115/14,440 patients with available information were included in the analysis of risk for hospitalization [14,18,24,26,28,35,36]. The prevalence of hospitalization tended to be higher during the first infection (12.8%, 1803/14,115) than the second infection (10.3%, 1452/14,115), with a pooled OR of 1.28 (95% CI: 1.19–1.37; <0.0001) and 0% heterogeneity (*p* = 0.59) (Figure 3). However, one study accounted for 97.6% of all pooled data [24].

A total of 11 studies reported information on transfer to the ICU for both episodes of infection in 1037 patients [13,14,15,16,17,18,20,23,26,28,34]. Of them, 984 patients from three studies with at least one event of transfer to the ICU were included in the analysis [26,28,34]. Transfer to the ICU tended to be lower during the first infection (0.71%, 7/984) than the second infection (1.52%, 15/984), with a pooled OR of 0.5 (95% CI: 0.20–1.25; *p* = 0.14) and 0% heterogeneity (*p* = 90.43) (Figure 4). However, two studies accounted for 90.8% of the data.

A total of 7/16,616 (0.4%) reinfected patients died, with information reported in 18 studies [13,14,15,16,17,18,20,21,22,23,24,26,28,30,31,34,35,37].

### 3.8. Risk Factors for Reinfection

Seven studies addressed this issue [22,24,32,33,34,37,38] (Appendix A). The lack of anti-nucleocapsid IgG in recovered patients [22], female gender [24,34], comorbidities [38], and unvaccinated status [33,34,37,38] were associated with a higher risk of reinfection. Contradictory results regarding age were found in two studies, with one reporting a higher risk of reinfection in younger patients [34] and another reporting an increased risk for reinfection in the oldest age categories (70–79 years old and 80+) [24]. A higher risk for reinfection was also observed during the Delta [38] and Omicron waves [32,34].

## 4. Discussion

An early systematic review, mostly based on Chinese data and describing 466 reinfection cases, concluded with an estimated 15% reinfection rate [39]. This high reinfection rate is probably due to the lack of a clear definition for reinfection, which likely unduly considered discharged patients with a second positive test as being “reinfected” [39]. We already discussed this issue in a previous paper, showing that PCR sensitivity and specificity and sampling methods likely account for an overestimation of the SARS-CoV-2 reinfection rate [40]. In a subsequent study based on 1096 reinfection cases across three continents, a more realistic pooled reinfection rate of 0.7% was found, in line with our 0.9% result on 18,175 reinfected patients on five continents [41]. However, due to the many epidemics of new SARS-CoV-2 variants that continue to occur worldwide, it is likely that the reinfection rate will slightly increase in future studies as a result of a cumulative effect. At our institute, we observed that the prevalence of reinfection among SARS-CoV-2 infection increased from 0.2% during the second epidemic to 6.8% during the fifth wave of epidemic due to the Omicron variant [4]. Most reinfected patients were middle-aged, which was in line with previous results [39]; we did not find significant differences in age between reinfected patients and those with only a primary reinfection, when data was available. Furthermore, data about age as a risk factor for reinfection in studies where it was specifically evaluated were contradictory. Our data suggests that studies conducted on healthcare workers show a higher rate of reinfection, which might be due both to a higher likelihood of testing in this population and a higher occupational risk of exposure to the virus.

The time between two episodes of infection was about five months, as compared to two months in a previous meta-analysis based on 577 cases of reinfection [42]; this disparity likely results from a mechanical effect due to the naturally increased duration of the pandemic, as new studies are conducted.

We confirm the results of colleagues regarding the highest prevalence of reinfection in female patients [42]. This observation may be a result of lower screening rates in males and higher occupational exposure among females.

We found no clear evidence of different clinical patterns between primary and second infection, although our analysis was conducted on a small sub-sample of only 168 patients [40].

Dhillon et al. reported a higher proportion of transfer to the ICU during reinfection [42], which differed from our findings. However, differences were not statistically significant in either study.

Data on vaccination status are too heterogeneous to allow strong conclusions about the effectiveness of vaccination in lowering reinfection rates, though some studies suggest a potential protective effect [20,30]. Nevertheless, one recent study suggests that vaccination has no effect on reinfection rate, regardless of the number of doses provided [43].

We acknowledge that our review has some limitations. Firstly, high heterogeneity was observed in all studies analyzed. These results might have been caused by various factors, including methodology, sample size and, especially, the periods when studies were conducted, given the prevalence different SARS-CoV-2 variants at different stages in the COVID-19 pandemic.

## 5. Conclusions

Reinfection with SARS-CoV-2 does occur, suggesting that natural immunity is not long-lasting in COVID-19 patients. Female patients seem more likely to experience reinfection than male patients. Thus far, no clear evidence for a difference in severity between the first infection and reinfection has been observed.

## Figures and Tables

**Figure 1 viruses-15-00967-f001:**
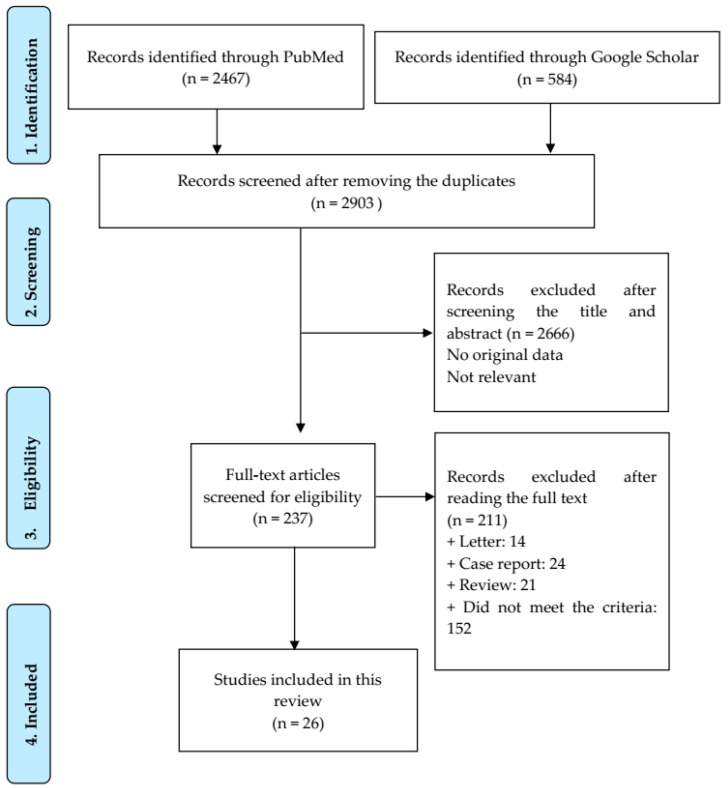
PRISMA flowchart of selected studies.

**Figure 2 viruses-15-00967-f002:**
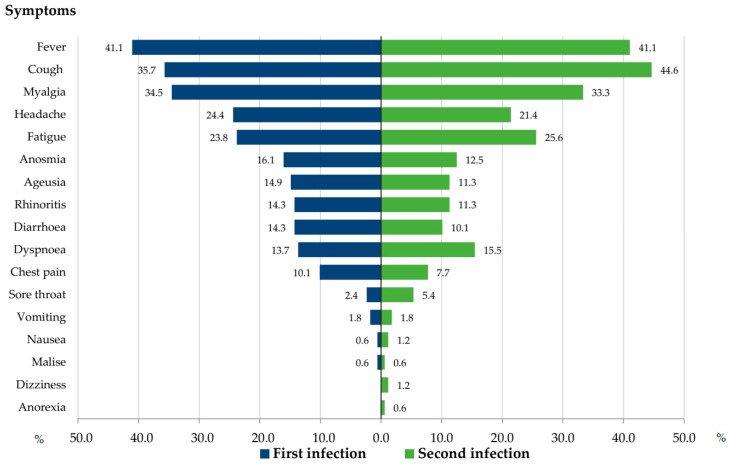
Percentage of patients with clinical symptoms at first and second infection.

**Figure 3 viruses-15-00967-f003:**
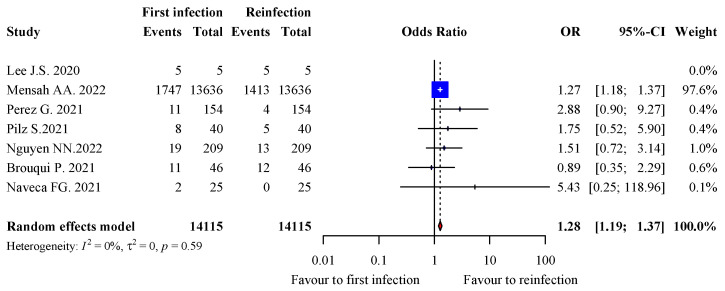
Forest plot of hospitalisations during the first and second infection.

**Figure 4 viruses-15-00967-f004:**
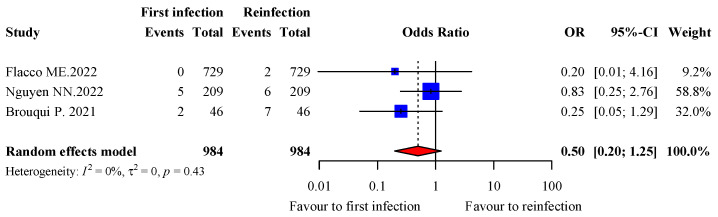
Forest plot of critical/severe disease during the first and second infection.

## Data Availability

Data are contained within the article.

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
