# Peer review of "SARS-CoV-2 Reinfection and Severity of the Disease: A Systematic Review and Meta-Analysis"

_viruses, 2023, doi:10.3390/v15040967_

Round 1

Reviewer 1 Report

This manuscript looks like a meta analysis of the previous work rather than a review. Therefore I suggest the authors submit this manuscript as a research work.

As a meta-analysis of the COVID-19 reinfections, there are far more literatures published. The authors selected only 23 studies. This could be improved by collecting more data.

The meta-analysis should be more thorough. Moreover, genetics should be emphasized in the reinfection studies. Is there significant trend of reinfections concerning the variants (delta, omicron,...) and substrains? Are such trends similar or different in different continent? Is there any association between human genetic background and vulnerability towards reinfection? Such information are of great interest and should be included in this manuscript.

Author Response

This manuscript looks like a meta analysis of the previous work rather than a review. Therefore I suggest the authors submit this manuscript as a research work.

Answer: Thank you very much for your comment, however, meta-analysis in the journal Viruses are published in the review section.

As a meta-analysis of the COVID-19 reinfections, there are far more literatures published. The authors selected only 23 studies. This could be improved by collecting more data. The meta-analysis should be more thorough.

Answer: Thank you for the comments. We updated the literature search until March 31, 2023 and included now 26 studies that meet the inclusion criteria. We agree that there are many studies addressing SARS-CoV-2 reinfection, but here, we choose to only include studies conducted in adults with reinfection cases confirmed by RT-PCR and / or genome sequencing. We excluded studies based on antigen testing and those with patients with a duration between two episodes <60 days when sequencing results were not available. This is stated in the material and method section.

Moreover, genetics should be emphasized in the reinfection studies. Is there significant trend of reinfections concerning the variants (delta, omicron,...) and substrains? Are such trends similar or different in different continent? Is there any association between human genetic background and vulnerability towards reinfection? Such information are of great interest and should be included in this manuscript.

Answer: To answer this question, we identified the most frequent circulating variant at the time of reinfection according to the period of study by using data from the CoVariants website. The results confirmed that the highest pooled proportion of reinfection was observed in patients reinfected with Omicron variant. We also observed a relatively high rate or reinfection with delta variant. This data was added to the draft in the result section and in the supplementary material (Supplementary Figure 3)

For human genetic background, we investigated the prevalence of reinfection by region which can be considered as a proxy for genetic human genetic background to some extent. No marked differences were seen between regions. This data was added to the draft in the result section and in the supplementary material (Supplementary Figure 2)

Reviewer 2 Report

In their systematic review, Nguyen et al. focus on SARS-CoV-2 reinfections. In sum, they analysed 23 studies including 18175 reinfected individuals and found an estimated reinfection rate ranging from 0.1 to 6%. They identified more reinfections during the omicron variant period and found females, individuals lacking specific IgG response after the first infection and unvaccinated being at higher risk for reinfections.

In general, the systematic review is well designed and search and inclusion strategies are described in an appropriate manner: The authors started with 2851 studies found in Pubmed and GoogleScholar and present a thoughtful PRISMA flowchart of finally 23 selected studies, explaining why studies were included and excluded. Interestingly, a very similar study design was used by the same authors in a review focusing on long-COVID that was published in 2022 in the journal European Journal of Clinical Microbiology & Infectious Diseases (https://doi.org/10.1007/s10096-022-04417-4).

Main points of criticism:

11. Table 1 contains a lot of information (which is good), but data presentation is rather poor and unclear. I strongly recommend revising the layout completely. You may also consider moving this table into the supplementary section, which might allow for a larger and more generous layout. This would help the reader to understand the results more quickly. Furthermore, the following points must be clarified:

·       One would expect finding two numbers in the column “Number of reinfection cases and vaccination status”. However, frequently there is only one number. The reader may guess whether this is the number of reinfections or the number of reinfected being vaccinated. The same holds for the next column providing information on discharged patients. I strongly recommend separating the information provided.

·       Other entries are confusing as well: As stated for study [26] (a self-citation by the way), it is stated “209, 0/207 patients vaccinated before the first infection (…)” – Do you mean 2 out of 207 patients were vaccinated? Is 209 the total number of individuals included?

·       The study period is of importance when it comes to the comparison of reinfections with different virus variants. However, instead of or even better in addition to the study date, you should provide information on the main virus variant circulating at that point in time at the location where the study was performed. John-Hopkins and/or official (governmental) websites offer information regarding this issue.  

·       Please check the content of the table once more for correctness: The study with reference number 34 (Pilz et al.,) was conducted in Austria (NOT Australia), thus your numbers regarding study locartions are not completely correct, as this study was conducted in Europe, not Asia/Australia.  

22. The results of the data analysis are described within the text, though sometimes it is a mere listing of what the reader has already seen/read in the table. Summarizing the information provided in the table in a few sentences is definitely a very good idea, however, I would strongly recommend revising this section and pool it with the information provided in the discussion section. Separating results and discussion section is not appropriate for a review, not even a systematic one.

3. You should also try to summarize the information found in a more detailed manner (e.g. regarding the average age of participants: the overall age from about 20 studies with about 4 million individuals ranges between 30 and 50 years (lines 165 ff). This would give the reader a rough idea about the information provided – easy and better to remember).

44. Please add meaningful subheadings for the results section (particularly for 3.3.), telling the reader the key message with a few words or a single sentence. That would greatly promote the readability of your manuscript.

5. Please add adequate figure legends (including supplementary figures) – not just a header.

Minor comments and typing errors:

·       Please check the use of punctuation throughout the whole manuscript. E.g. line 136, line 181, line 334; lines 335-340: use dots or do not, but please do not mix it up.

·       Please check the use of spaces in Figure 1: there is almost for every n-number given a different use of spaces. In addition, please stick to one expression for n-numbers. In figure 1 you say “n =…”, while throughout the text you use “N =…” (line 140 ff).

·       Line 253 and line 265: it should be first and second infection (not infections).

·       Line 108: Again, this is Austria and NOT (!) Australia. I am using R myself quite frequently and I know this particular citation quite well.

·       Line 236: Missing number or one too many? 14115 out of 1440 patients?

Author Response

Reviewer 2:

In their systematic review, Nguyen et al. focus on SARS-CoV-2 reinfections. In sum, they analysed 23 studies including 18175 reinfected individuals and found an estimated reinfection rate ranging from 0.1 to 6%. They identified more reinfections during the omicron variant period and found females, individuals lacking specific IgG response after the first infection and unvaccinated being at higher risk for reinfections.

In general, the systematic review is well designed and search and inclusion strategies are described in an appropriate manner: The authors started with 2851 studies found in Pubmed and GoogleScholar and present a thoughtful PRISMA flowchart of finally 23 selected studies, explaining why studies were included and excluded. Interestingly, a very similar study design was used by the same authors in a review focusing on long-COVID that was published in 2022 in the journal European Journal of Clinical Microbiology & Infectious Diseases (https://doi.org/10.1007/s10096-022-04417-4).

Answer: Thank you very much for your comment

Main points of criticism:

  1. Table 1 contains a lot of information (which is good), but data presentation is rather poor and unclear. I strongly recommend revising the layout completely. You may also consider moving this table into the supplementary section, which might allow for a larger and more generous layout. This would help the reader to understand the results more quickly. Furthermore, the following points must be clarified:

Answer: Thank you for all the valuable comments. We moved the table the supplementary files and revised the layout (see below).

  One would expect finding two numbers in the column “Number of reinfection cases and vaccination status”. However, frequently there is only one number. The reader may guess whether this is the number of reinfections or the number of reinfected being vaccinated. The same holds for the next column providing information on discharged patients. I strongly recommend separating the information provided.

Answer: We separated the information into two different columns as request (Supplementary Table 1)

  • Other entries are confusing as well: As stated for study [26] (a self-citation by the way), it is stated “209, 0/207 patients vaccinated before the first infection (…)” – Do you mean 2 out of 207 patients were vaccinated? Is 209 the total number of individuals included?

Answer: this has been corrected

  • The study period is of importance when it comes to the comparison of reinfections with different virus variants. However, instead of or even better in addition to the study date, you should provide information on the main virus variant circulating at that point in time at the location where the study was performed. John-Hopkins and/or official (governmental) websites offer information regarding this issue.  

Answer: Thanks for your comments. We added the information in the column “Most frequent variant circulation at the time of reinfection in the area” as recommended (Supplemenrary Table 1)

  • Please check the content of the table once more for correctness: The study with reference number 34 (Pilz et al.,) was conducted in Austria (NOT Australia), thus your numbers regarding study locations are not completely correct, as this study was conducted in Europe, not Asia/Australia.  

Answer: We corrected this information as required  

  1. The results of the data analysis are described within the text, though sometimes it is a mere listing of what the reader has already seen/read in the table. Summarizing the information provided in the table in a few sentences is definitely a very good idea, however, I would strongly recommend revising this section and pool it with the information provided in the discussion section. Separating results and discussion section is not appropriate for a review, not even a systematic one.

Answer: We have considered your suggestion, but according to Viruses journal style, the two sections should be separated. In the result section, we present the performed analysis and we highlight the most important result for the reader. Details can be found in the large supplementary table.

  1. You should also try to summarize the information found in a more detailed manner (e.g. regarding the average age of participants: the overall age from about 20 studies with about 4 million individuals ranges between 30 and 50 years (lines 165 ff). This would give the reader a rough idea about the information provided – easy and better to remember).

 Answer:  The information was modified as request and this paragraph was shortened. The gender paragraph was also shortened.

  1. Please add meaningful subheadings for the results section (particularly for 3.3.), telling the reader the key message with a few words or a single sentence. That would greatly promote the readability of your manuscript.

 Answer: We added subheading, however, as per Vaccine journal style, we kept generic subheadings instead of sentences with key message

  1. Please add adequate figure legends (including supplementary figures) – not just a header.

Answer: Figure titles were revised and legend added when necessary

Minor comments and typing errors:

  • Please check the use of punctuation throughout the whole manuscript. E.g. line 136, line 181, line 334; lines 335-340: use dots or do not, but please do not mix it up.

Answer: punctuation was checked throughout the manuscript

  • Please check the use of spaces in Figure 1: there is almost for every n-number given a different use of spaces. In addition, please stick to one expression for n-numbers. In figure 1 you say “n =…”, while throughout the text you use “N =…” (line 140 ff).

Answer: spaces were checked and n was used throughout the manuscript

  • Line 253 and line 265: it should be first and second infection (not infections).

Answer: this was corrected

  • Line 108: Again, this is Austria and NOT (!) Australia. I am using R myself quite frequently and I know this particular citation quite well.

Answer: This was corrected

  • Line 236: Missing number or one too many? 14115 out of 1440 patients?

Answer : This was corrected (14115/14440) (line 279)

Reviewer 3 Report

The authors have summarized the results of 23 studies addressing SARS-CoV-2 reinfections in this manuscript.

Several suggestions:

1.      In Lines 2 and 142, it is suggested to use [SARS-CoV-2] rather than [COVID-19].

2.      In Lines 29 and 270, please change [anti-nucleoside IgG] to [anti-nucleocapsid IgG].

3.      Line 181, [.] is missing between [[21,26,32]] and [Only].

4.      Line 291, is [fifth one] means [fifth wave of epidemic]?

5.      Line 308, reference 37 is suggested to add after [only 168 patients].

6.      Line 321. If [natural immunity is not long-lasting in COVID-19 patients] is the conclusion, please mention the status of immunity after the first infection of these re-infected patients. For example, the titer of anti-nucleocapsid IgG in reference 22, or [even better] the titer of anti-spike IgG.

7.      I believe that [two distinct virus strains cause re-infection] in reference 2 and [Genome sequences showed that genetically distinct SARS-CoV-2 strains were responsible for reinfections.] in reference 40. Thus, please mention the different virus strains causing the re-infection in these 23 studies.

Author Response

The authors have summarized the results of 23 studies addressing SARS-CoV-2 reinfections in this manuscript.

Several suggestions:

  1. In Lines 2 and 142, it is suggested to use [SARS-CoV-2] rather than [COVID-19].

Answer: This was corrected as recommended

  1. In Lines 29 and 270, please change [anti-nucleoside IgG] to [anti-nucleocapsid IgG].

Answer: The word was corrected as suggested

  1. Line 181, [.] is missing between [[21,26,32]] and [Only].

Answer: This sentence was corrected as suggested

  1. Line 291, is [fifth one] means [fifth wave of epidemic]?

Answer: Yes, it is. This was corrected

  1. Line 308, reference 37 is suggested to add after [only 168 patients].

Answer: The references was added as recommended (it changed to reference 40).

  1. Line 321. If [natural immunity is not long-lasting in COVID-19 patients] is the conclusion, please mention the status of immunity after the first infection of these re-infected patients. For example, the titer of anti-nucleocapsid IgG in reference 22, or [even better] the titer of anti-spike IgG.

Answer: The information was added to result section and supplementary table 1

  1. I believe that [two distinct virus strains cause re-infection] in reference 2 and [Genome sequences showed that genetically distinct SARS-CoV-2 strains were responsible for reinfections.] in reference 40. Thus, please mention the different virus strains causing the re-infection in these 23 studies

Answer: The information on variants when available was added as recommended in the result section and in supplementary table 4)

Round 2

Reviewer 2 Report

Dear authors,

Thank you for answering point-by-point, I have no further comments or suggestions.